# Modeling Climate Change Indicates Potential Shifts in the Global Distribution of Orchardgrass

Jiqiang Wu [1,†], Lijun Yan [2,†], Junming Zhao [1], Jinghan Peng [1], Yi Xiong [1], Yanli Xiong [1] and Xiao Ma [1,*]

[1] College of Grassland Science and Technology, Sichuan Agricultural University, Chengdu 611130, China; immortalwjq@outlook.com (J.W.); junmingzhao163@163.com (J.Z.); m17623220313@163.com (J.P.); xiongyi95@126.com (Y.X.); yanlimaster@126.com (Y.X.)

[2] Sichuan Academy of Grassland Sciences, Chengdu 611743, China; yanlijun456@126.com

[*] Correspondence: maxiao@sicau.edu.cn

[†] These authors contributed equally to this work and should be considered co-first authors.

**Abstract:** Orchardgrass (*Dactylis glomerata* L.) is highly tolerant of shade, cold, and overwintering, making it an ideal species for grassland ecological restoration and livestock production. However, the genetic diversity of orchardgrass may be threatened by climate change. Using a Maximum Entropy (MaxEnt) model with the BCC-CSM2-MR global climate database and the Harmonized World Soil Database, we projected the current and future distribution of orchardgrass suitable areas globally. The predicted ecological thresholds for vital environmental factors were determined to be a temperature seasonality range of 411.50–1034.37 °C, a mean diurnal range of −0.88–10.69 °C, a maximum temperature of the warmest month of 22.21–35.45 °C, and precipitation of the coldest quarter of 116.56–825.40 mm. A range of AUC values from 0.914 to 0.922, indicating the accuracy of the prediction model. Our results indicate that the total area of current suitable habitats for orchardgrass was estimated to be 2133.01 × 10⁴ km², it is dispersed unevenly over six continents. Additionally, the suitable areas of habitats increased in higher latitudes while decreasing in lower latitudes as greenhouse gas emissions increased. Therefore, efforts should be made to save places in the southern hemisphere that are in danger of becoming unsuitable, with the possibility of using northern America, China, and Europe in the future for conservation and extensive farming.

**Keywords:** bioclimatic; climate change; *Dactylis glomerata*; habitat shift; MaxEnt model; potential geographic distribution

## 1. Introduction

Orchardgrass (*Dactylis glomerata* L.) is a perennial cold-season forage grass of the Poaceae family native to northern Africa, Europe, and some temperate regions of Asia [1,2]. It plays a crucial role in the generation of forage-based meat and dairy across temperate areas as one of the top four commercially significant forage types of grass grown globally [3,4]. Orchardgrass is known for its rapid growth, high biomass, high sugar content, strong shade tolerance, and adaptability to different environments [5–8]. Additionally, orchardgrass has been cultivated in North America for over two centuries and is currently one of the most widely cultivated grass, primarily for grass grazing and hay production [1]. Orchardgrass exhibits high genetic diversity, extensive geographical distribution, and varied habitat conditions, making it an excellent candidate grass for further genetic and ecological studies [2]. The temperature, moisture, and soil conditions are key factors influencing the growth and development of vegetation in orchardgrass [9–11]. Due to climate change, the regional climate patterns might change, and catastrophic climatic phenomena such as heat and droughts will become more frequent, leading to the extinction of species that cannot adapt to the environment or have a limited capacity to adjust [12]. Climate change will likely cause further harm to orchardgrass varieties and habitats in the future. Therefore, it is vital to improve the management and preservation of the main distribution regions of

orchardgrass and safeguard critically threatened natural populations through in situ and ex-situ conservation measures.

Niches are habitats with the minimum thresholds necessary for survival [13]. The grassland niche is highly influenced by the surrounding environment, causing it to adapt or relocate in response to environmental changes. The Earth's temperature is projected to rise by 0.2 °C per decade due to greenhouse gas (GHG) emissions, with expected increases ranging from 0.3–1.7 °C at a minimum, up to 2.6–4.8 °C at a maximum for the twenty-first century [14]. As a widely distributed temperate perennial grass species, orchardgrass will inevitably struggle to move and colonize suitable habitats at a fast enough pace to cope with the predicted rapid climate change. This could result in elevated rates of species extinction, as well as diminished plant growth and yield due to the warmer temperatures [15,16]. As a result, there is a pressing need to determine the amount of climate change during the next few decades and evaluate its impact on specific indigenous grass habitats using a variety of techniques, which will develop future conservation and cultivation plans [17].

Species distribution modeling is a developing field of study that employs niche theory to derive the ecological requirements of specific species using mathematical models. These models combine environmental factors and occurrence data to provide a statistical or mechanical representation of the organism's probable distribution [18–20]. Currently, the most widely utilized niche models for species distribution are GARP (Genetic Algorithm for Rule-set Production) [21], MaxEnt (Maximum Entropy modeling) [22], Bioclim [23], Random Forest [24], and the Boosted Regression Tree [25]. Notably, MaxEnt, in accordance with the principle of maximum entropy [19,22,26], is frequently regarded as outperforming other species distribution models (SDMs) due to its strong toleration and precise forecasting in many model intercomparisons [27–29]. Researchers worldwide in the last decade have achieved significant success in applying species distribution models to issues such as protecting the diversity of rare animals and plants [30–33], estimating the dangers of invasive species [34–36], protecting marine ecosystem [37,38], predicting disaster distribution [39], and disease propagation [40,41], employing the MaxEnt model.

In this study, we used the MaxEnt model, combined with climatic factors, terrain factors, and soil factors, to predict the suitable area for orchardgrass. The four goals of this study were to: (1) assess the key external variables affecting the distribution of orchardgrass; (2) investigate the distribution of suitable areas for orchardgrass under present and future climate scenarios; (3) predict potential distribution shifts of orchardgrass; (4) pinpoint areas of habitat expansion and degradation for orchardgrass.

## 2. Materials and Methods

### 2.1. Data on Species Occurrence

To collect extensive data on the global natural distribution of orchardgrass, we searched multiple databases, including the Global Biodiversity Information Facility (GBIF; https://www.gbif.org/ (accessed on 13 May 2023)) and the Chinese Virtual Herbarium (CVH; https://www.cvh.ac.cn/ (accessed on 13 May 2023)) [42]. Furthermore, the scientific names of our target species (*Dactylis glomerata*) were used as search terms in a Web of Science (WOS; https://www.webofscience.com (accessed on 13 May 2023)) database search, and we recorded all the distribution sites that were mentioned in the literature. We conducted searches in the Chinese National Knowledge Infrastructure (CNKI; https://www.cnki.net/ (accessed on 13 May 2023)) database using the Chinese and scientific names of the target species [42]. This database is one of the most comprehensive databases in the Chinese scientific field and also records the distribution points of the target species. By utilizing these resources, we compiled a comprehensive list of orchardgrass occurrence records and a distribution map. (Figure 1; Table S1).

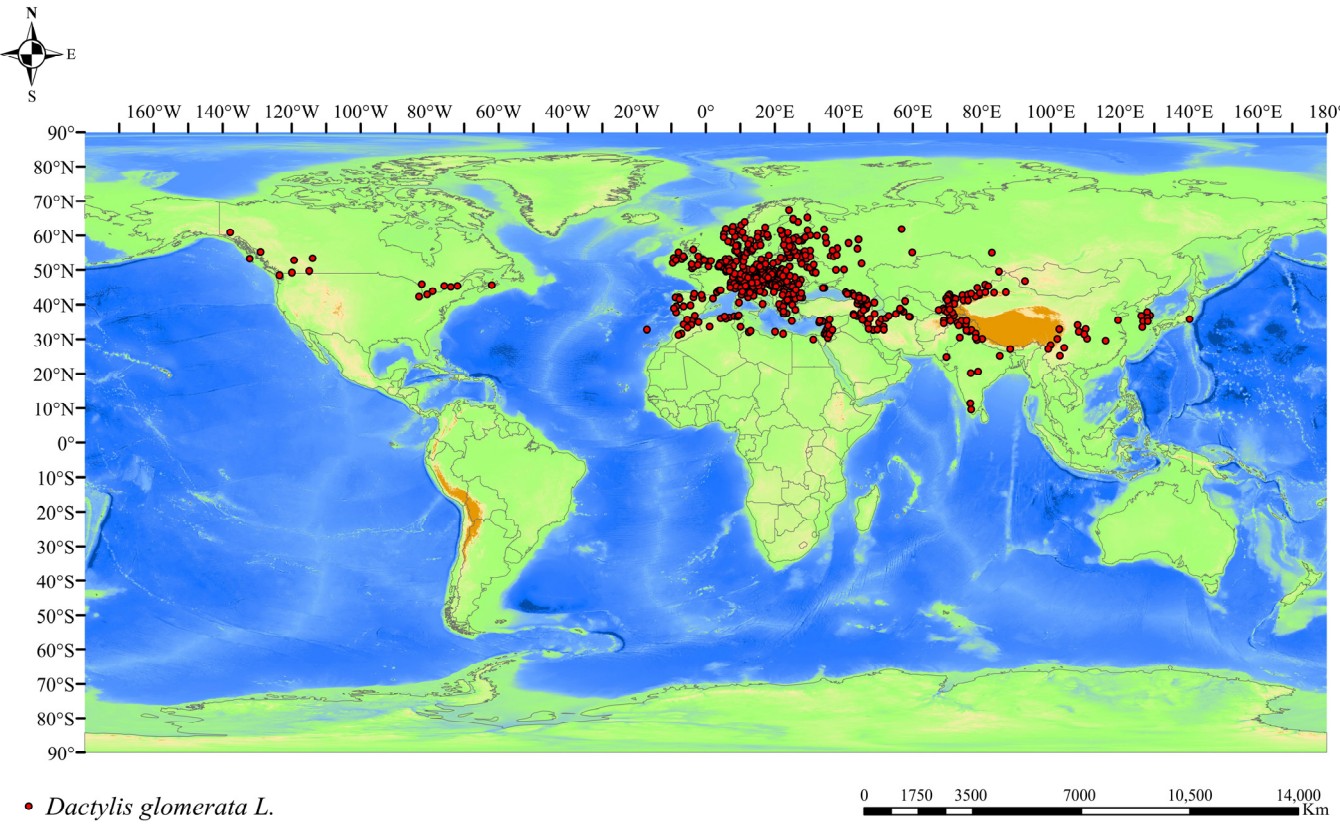

• *Dactylis glomerata L.*

**Figure 1.** Records points of orchardgrass occurrence created in ESRI ArcMap 10.4.1.

The distribution data of our target species (orchardgrass) underwent several processing steps. Firstly, we convert the longitude and latitude values from sexagesimal to decimal. Secondly, we individually verified whether the reported latitudes and longitudes matched the corresponding collecting locations. In cases where the records did not match, we removed them from the dataset. We then implemented the ENMTools.pl version 1.0.4 software (https://github.com/danlwarren/ENMTools (accessed on 13 May 2023)) to reduce the impact of sampling bias by trimming the occurrence points to retain only one observation per 5 arc-min grid cell. The environmental data are the same for each grid cell [43]. In doing so, we obtained 522 orchardgrass occurrence points.

*2.2. Environmental Variables*

Climate data and terrain data for this study were obtained from the WorldClim database (https://www.worldclim.org/ (accessed on 13 May 2023)) and soil data were obtained from the Harmonised World Soil Database (https://www.fao.org/soils-portal/soil-survey/soil-maps-and-databases/harmified-world-soil-database-v12/ (accessed on 13 May 2023)) [44]. In total, 19 bioclimatic variables, 10 soil variables, and 1 terrain variable (Table 1) were used as candidates for constructing the MaxEnt model. The climatic data from the years 1970 to 2000 were used to depict the current climate, while the predicted future climate data came from the most recent WorldClim version 2.1 (https://www.worldclim.org/ (accessed on 13 May 2023)). Under the BCC-CSM2-MR climate model, we forecast alterations in suitable areas for the time periods 2041–2060 and 2081–2100 using the average values of three Shared Socio-economic Pathways (SSPs) [45]. The SSPs, which are anticipated to represent various ranges of GHG emission levels, included SSP126 (low GHG emissions), SSP370 (mid-GHG emissions), and SSP585 (high GHG emissions). In all, this study used 30 different environmental factors, such as soil type, altitude, and climate. We used the SDMtools resampling and transformation tool in ArcGIS (version 10.4.1) to resample the environmental data. The data was converted to a 5′ resolution and saved in

.asc format [46]. The resampling technique employed was nearest neighbor resampling, where the nearest pixel value from the original data was selected as the new pixel value to ensure data quality and accuracy.

**Table 1.** Meaning and units of environment variables used in this study. The variables in bold are the sixteen variables remaining after an iterative variable reduction process and are the variables used in the final climate envelope model.

| Index | Variable | Source | Description | Unit |
|---|---|---|---|---|
| Bioclimatic variables | BIO1 | Worldclim | Annual Mean Temperature | °C |
| | **BIO2** | | **Mean diurnal range (mean of monthly (max temp–min temp))** | **°C** |
| | BIO3 | | Isothermality (BIO2/BIO7) (×100) | % |
| | **BIO4** | | **Temperature seasonality (standard deviation × 100)** | **°C** |
| | **BIO5** | | **Max temperature of the warmest month** | **°C** |
| | BIO6 | | Min temperature of the coldest month | °C |
| | BIO7 | | Temperature annual range (BIO5–BIO6) | °C |
| | BIO8 | | Mean temperature of wettest quarter | °C |
| | BIO9 | | Mean temperature of driest quarter | °C |
| | BIO10 | | Mean temperature of warmest quarter | °C |
| | BIO11 | | Mean temperature of coldest quarter | °C |
| | BIO12 | | Annual precipitation | mm |
| | BIO13 | | Precipitation of wettest month | mm |
| | BIO14 | | Precipitation of the driest month | mm |
| | **BIO15** | | **Precipitation seasonality (coefficient of variation)** | **mm** |
| | **BIO16** | | **Precipitation of the wettest quarter** | **mm** |
| | **BIO17** | | **Precipitation of the driest quarter** | **mm** |
| | BIO18 | | Precipitation of the warmest quarter | mm |
| | **BIO19** | | **Precipitation of the coldest quarter** | **mm** |
| Terrain variables | **Elevation** | Worldclim | **Elevation** | **m** |
| Soil variables | **ESP** | Harmonised World Soil Database | **Exchangeable sodium percentage** | **—** |
| | **Gravel** | | **Volume percentage of gravel** | **—** |
| | **OC** | | **Percentage of organic carbon** | **—** |
| | **PH** | | **Soil reaction** | **mol·L$^{-1}$** |
| | **AWC** | | **Available water capacity** | **g/kg** |
| | **Bulk** | | **Cation exchange capacity** | **cmol (+)/kg** |
| | **Clay** | | **Percentage of clay** | **—** |
| | **Drainage** | | **Soil drainage class** | **—** |
| | CECS | | Cation exchange capacity of the clay fraction | — |
| | Sand | | Percentage of sand | — |

ENMTools.pl was utilized to examine the correlation between the candidate variables (19 bioclimatic variables, 1 terrain variable, and 10 soil variables) in order to prevent multi-collinearity among the variable inputs. The threshold of Pearson correlation coefficient was set at 0.75, and the correlated variables were removed accordingly [47]. Finally, we chose BIO2 (mean diurnal range), BIO4 (temperature seasonality), BIO5 (max temperature of the warmest month), BIO15 (precipitation seasonality), BIO16 (precipitation of wettest quarter), BIO17 (precipitation of driest quarter), BIO19 (precipitation of coldest quarter), ESP (Exchangeable sodium percentage), Gravel (Volume percentage of gravel), OC (percentage of organic carbon), AWC (available water capacity), Bulk (cation exchange capacity), Clay (percentage of clay), Drainage (soil drainage class), elevation, and PH, as the environmental factors to construct the model of species distribution (Figure 2, Table S2).

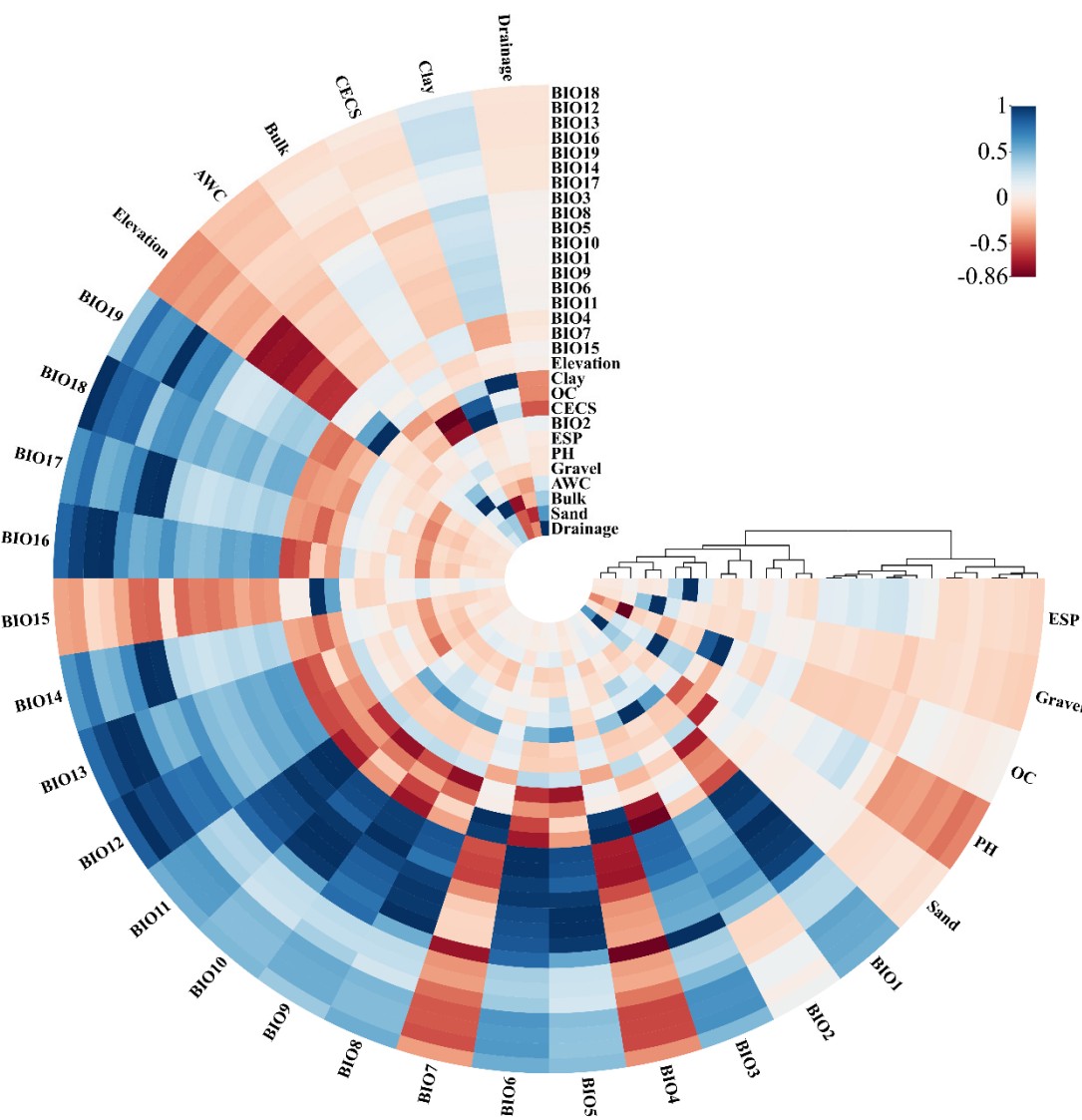

**Figure 2.** Pearson correlation coefficient of environmental variables.

## 2.3. Optimization of Model Parameters

We optimized the regularization multiplier and feature class parameters in R version 3.6.3 [48] using the Kuenm package (https://github.com/marlonecobos/kuenm (accessed on 13 May 2023)). This package is a prerequisite for building the species distribution model with MaxEnt version 3.4.4 (https://biodiversityinformatics.amnh.org/open_source/maxent/ (accessed on 13 May 2023)). The training set consisted of 75% of the data, which comprised of 392 occurrence points. It was determined that 1160 candidate models could be tested, representing all possible combinations of the 29 feature class combinations and the 40 regularization multiplier settings (0.1 to 4 with an interval of 0.1). The model's statistical significance, predictive performance, and complexity were evaluated in a sequential manner, with the partial ROC being assessed first, followed by the omission rates, and finally the AICc values. The statistical significance of the candidate models was the initial criterion for screening them. Second, to narrow the pool of models, omission rate requirements (i.e., 5% where feasible) were applied. In the final round of model selection, we chose models that had the lowest delta AICc values (<2) and met both the criteria of statistical significance and low omission rates.

*2.4. Species Distribution Model*

Construct species distribution models with MaxEnt 3.4.4 utilizing the above 16 environmental layers. Notably, the majority of earlier studies have concentrated on predicting suitable habitats in countries such as China and Pakistan using models that essentially use the standard parameters, leading to subpar model accuracy [49,50]. Therefore, to improve the accuracy of the model data, we employed the reg_mult function from the Kuenm package (https://github.com/marlonecobos/kuenm (accessed on 13 May 2023)). for model performance screening in our study. In addition to simulating regularization multipliers ranging from 0.1 to 4, the models with five types of feature categories (Hinge, Threshold, Linear, Quadratic, Product, and Threshold) also optimized their respective parameters [48,51]. Ten replicates were conducted for our analysis, with a maximum of 5000 iterations and 10,000 background points, encompassing the sixteen environmental layers mentioned earlier. The area under the receiver operating characteristic curve (AUC) was used to evaluate the model's performance [52]. An AUC value of 0.9–1.0 indicates a perfect prediction and a value of < 0.5 represents a random prediction [44]. Using the Jackknife test, we evaluated the relative contributions of each environmental variable [22]. Each grid cell has a probability of the model's output value being between 0 and 1, which can be read as a measure of relative suitability. The model outputs were categorized using the MTSPS (maximum test sensitivity plus specificity) threshold: grid cells with suitabilities above the MTSPS threshold were classified as potentially suitable habitat, while others were classified as unsuitable habitat [53]. We employed ArcGIS version 10.4.1 to create distribution maps, compute the percentage of potentially appropriate areas, and determine the average output of the BCC-CSM2-MR global model for the same SSP and identical period. The present study mainly summarizes the predicted mean results of the BCC-CSM2-MR global model under the different GHG emission scenarios in 2041–2060 and 2081–2100. The proportions of contracted and extended potentially suitable areas in orchardgrass, as well as changes in the potentially suitable areas of their centroids, were examined using SDMtoolbox version 2.4 software [54–56].

*2.5. Estimation of Orchardgrass Distribution Area*

In this study, we retrieved global land area data from the Food and Agriculture Organization of the United Nations (FAO) website (https://www.fao.org/ (accessed on 13 May 2023)) and used ArcGIS version 10.4.1 to divide it into suitable and unsuitable areas using the MTSPS method. By calculating the percentage of suitable and unsuitable grid cells out of the total grid cells and multiplying it by the corresponding area of each region, we can determine the area of suitable and unsuitable areas.

## 3. Results

### 3.1. Modeling of Species Distribution

The assessment revealed that the MaxEnt model with QPH (quadratic, product, and hinge) feature type parameters and a regularization multiplier of 2.6 was the most effective one that was used to proceed with further analysis. Under the current scenario, test AUC values from the final models' tenfold cross-validation are 0.919 (Figure 3), while the MaxEnt model's forecasts for potential orchardgrass habitats delivered positive outcomes, with a range of AUC values from 0.914 to 0.922 (Figure S1).

Internal jackknife testing of the MaxEnt model showed that temperature seasonality (Bio4, 34.9% of contribution), mean diurnal range (Bio2, 22.9% of contribution), the max temperature of the warmest month (Bio5, 7.8% of contribution), precipitation of coldest quarter (Bio19, 6.6% of contribution), contributed most to the Maxent model for orchardgrass, with an overall contribution of 72.2% (Table 2). Precipitation of wettest quarter (Bio16), PH, precipitation seasonality (Bio15), and other indicators made up 27.8% of the total contribution. Ecological thresholds for important environmental factors are known from environmental factor response curves (Figure S2): temperature seasonality (411.50–1034.37 °C), mean

diurnal range ($-0.88$–10.69 °C), precipitation of the coldest quarter (116.56–825.40 mm), and max temperature of the warmest month (17.08–40.84 °C).

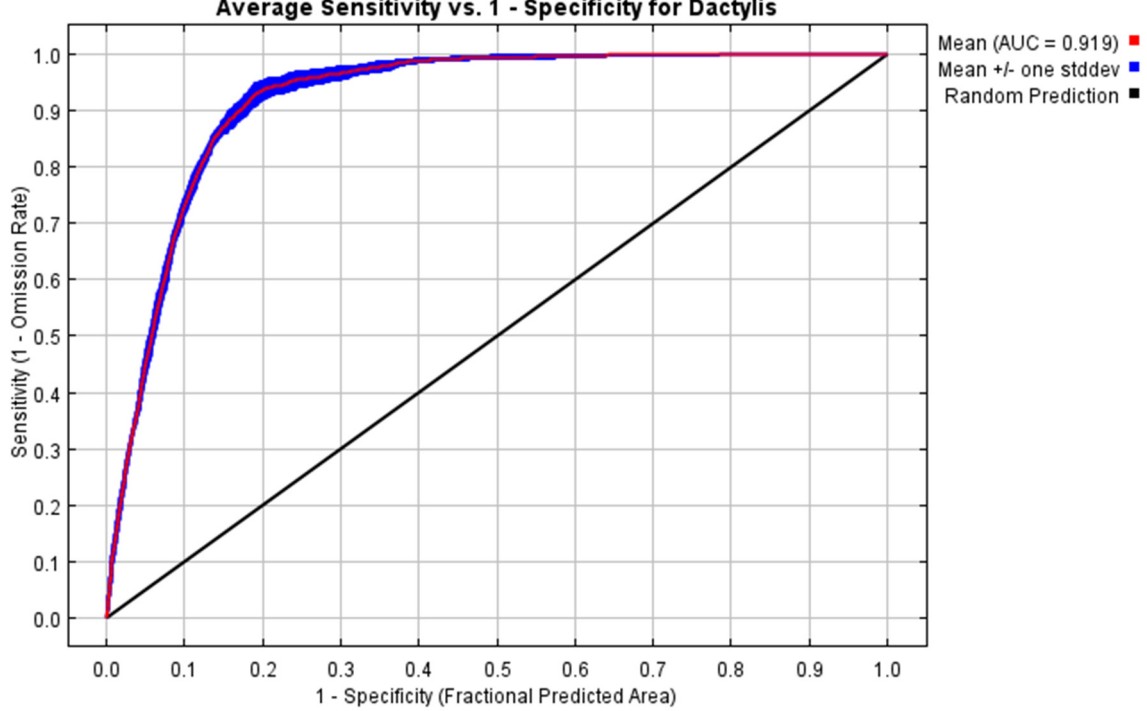

**Figure 3.** The final model of the receiver operating characteristic curve.

**Table 2.** Environmental factors contributing to the current suitable habitat for orchardgrass and their thresholds in MaxEnt models. Only the top four contributing environmental factors out of 16 are listed.

| Code | Environmental Factor | Percent Contribution (%) | Suitable Threshold | Units |
|------|---------------------|--------------------------|--------------------|-------|
| Bio4 | temperature seasonality | 34.9 | 411.50–1034.37 | °C |
| Bio2 | mean diurnal range | 22.9 | $-0.88$–10.69 | °C |
| Bio5 | max temperature of the warmest month | 7.8 | 17.08–40.84 | °C |
| Bio19 | precipitation of the coldest quarter | 6.6 | 116.56–825.40 | mm |

*3.2. Current Suitable Distribution for Orchardgrass*

Habitats for orchardgrass (Figure 4a) were widely distributed over three continents, primarily in western and southern North America, Europe, southwestern and southeastern Asia, and a few in southern South America, northern Africa, and southeastern Oceania, but not in Antarctica. The total appropriate habitat area was 2133.01 $\times$ 10$^4$ km$^2$, accounting for 14.19% of the world's land area.

Six continents had different suitable habitat distributions for orchardgrass (Figure 4b). Europe had the largest total area of appropriate habitat (754.79 $\times$ 10$^4$ km$^2$), covering most of the continent. Asia had the second biggest area, with a total appropriate habitat area of 597.64 $\times$ 10$^4$ km$^2$, largely in China, Turkey, and Iran. The appropriate habitat area for orchardgrass in North America was 247.80 $\times$ 10$^4$ km$^2$, largely found in coastal areas of the United States and Canada. Oceania has the smallest amount of suitable habitat for orchardgrass out of all the continents, with only 70.21 $\times$ 10$^4$ km$^2$ concentrated in the coastal areas of New Zealand and southeastern Australia. These findings support the current distribution pattern of orchardgrass and demonstrate the accuracy of using MaxEnt to predict species distribution patterns.

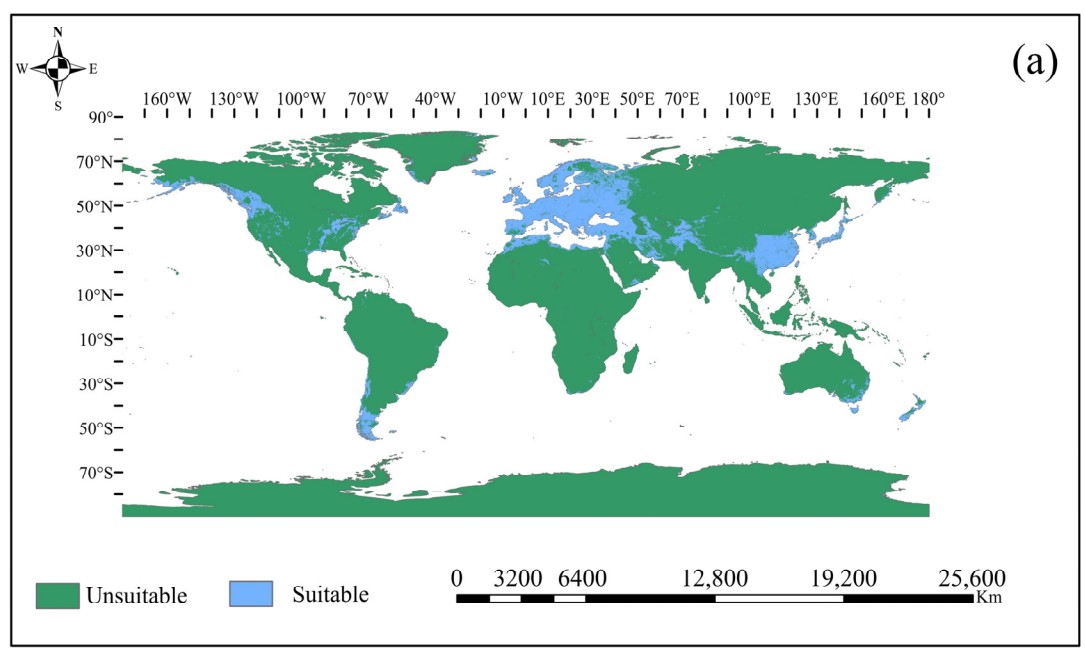

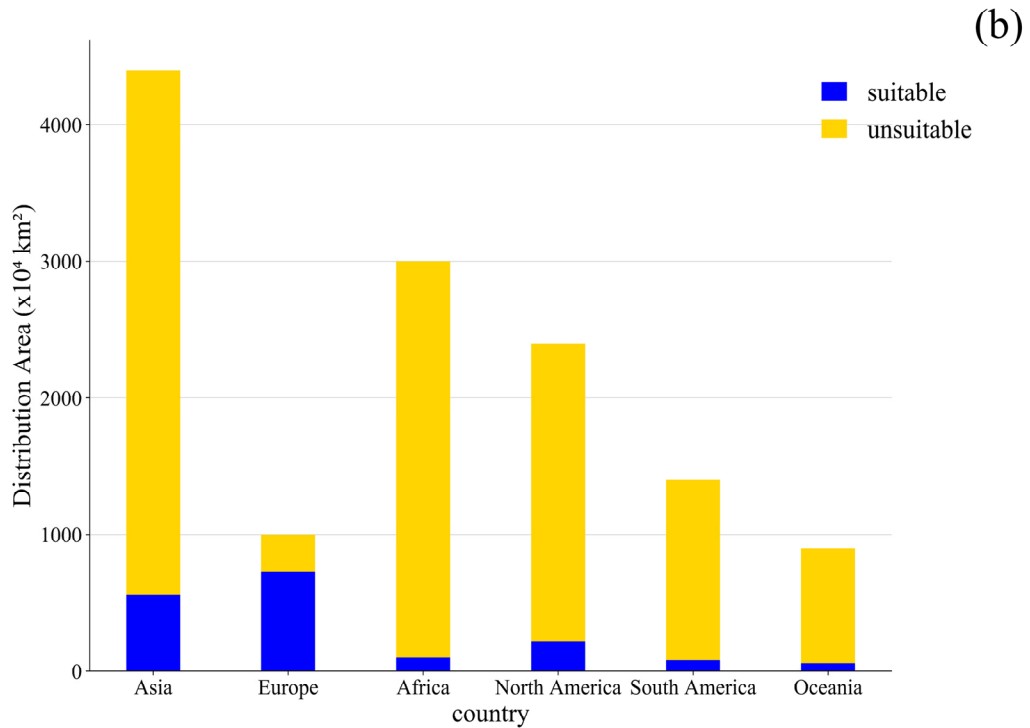

**Figure 4.** In accordance with current climate conditions, map (**a**) shows suitable habitat areas for orchardgrass. Created in ESRI ArcMap 10.4.1 map (**b**) shows a distribution area of orchardgrass habitats on six continents.

### 3.3. Potential Distribution of Orchardgrass under Future Climate Conditions

The study investigates the potential redistribution of orchardgrass habitats in the twenty-first century in response to climate change under three different scenarios. Based on a comparison of the present appropriate habitats (Figure 4a) with the predicted appropriate habitats for 2041–2060 and 2081–2100, we examined several tendencies that appeared under various climatic scenarios. (Figure 5). Total suitable habitat expanded from $75.91 \times 10^4$ km² (ssp245, 2041–2060) to $322.03 \times 10^4$ km² (ssp585, 2081–2100), while the contraction area

varied from 182.04 $\times$ 10$^4$ km2 (ssp126, 2041–2060) to 378.50 $\times$ 10$^4$ km$^2$(ssp585, 2081–2100). Overall, the findings demonstrated that rising GHG emissions were significantly associated with a global shrinkage in the extent of appropriate habitats for orchardgrass. During the second half of the twenty-first century, the southern hemisphere, more notably South America, Central Africa, and Oceania had a particularly dramatic decline. The shrinking of Asia and South America is the most noteworthy at the end of the twenty-first century, owing to increasing GHG emissions.

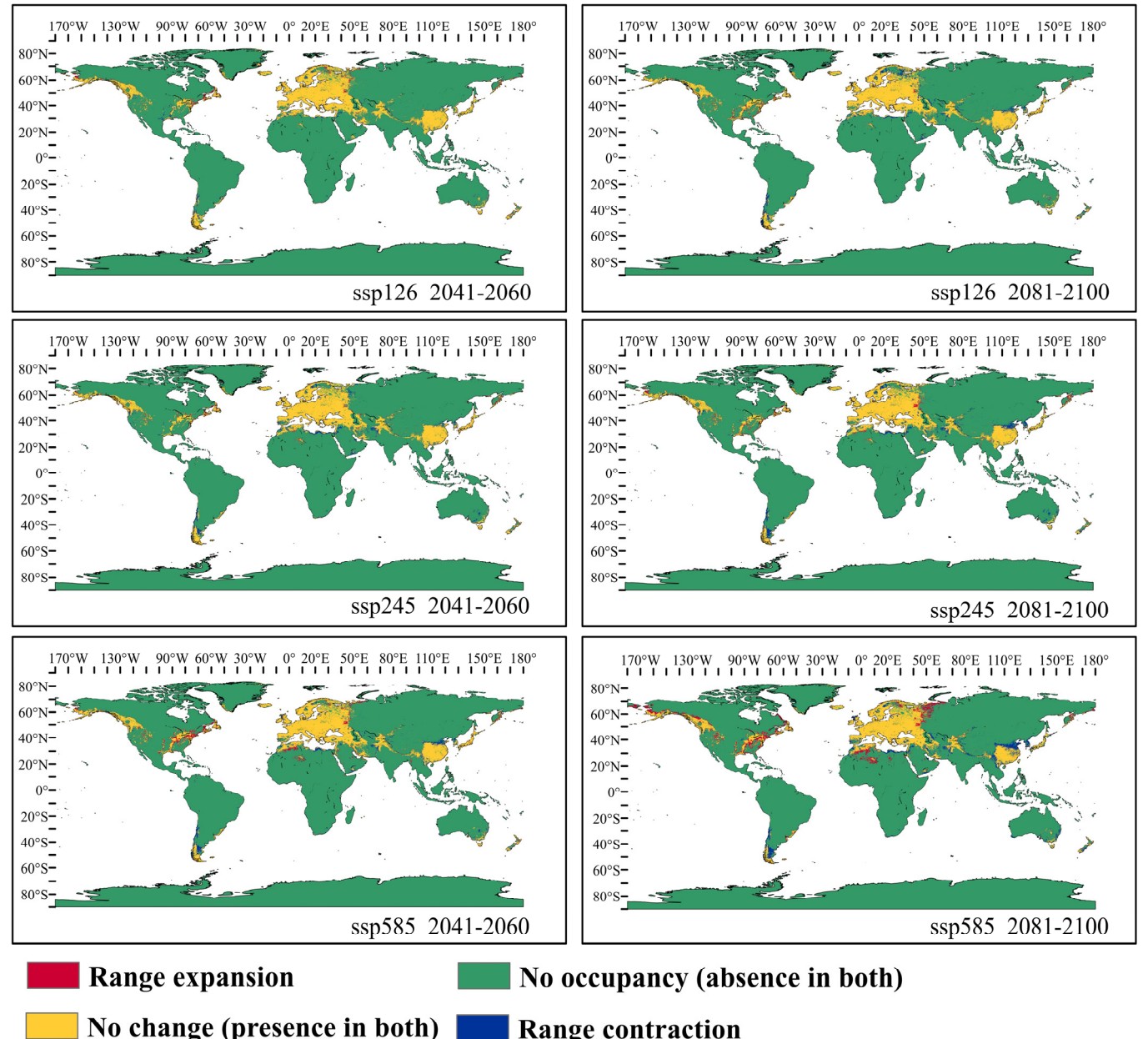

**Figure 5.** From years 2041 to 2100, changes in suitable habitats for orchardgrass under ssp126, ssp245, and ssp585 scenarios compared to the current distribution.

Significant expansion and contraction of suitable orchardgrass habitat towards the northwest were noted in all future climate scenarios, with significant differences between continents (Figure 6a, Table S3). Interestingly, appropriate growth regions in Asia, Europe, and North America stretch to high latitudes, with Asian habitat expansion ranging from 20.49 $\times$ 10$^4$ km$^2$ (ssp126, 2081–2100) to 38.13 $\times$ 10$^4$ km$^2$ (ss585, 2041–2060). European habitat expansion ranged from 6.24 $\times$ 10$^4$ km$^2$ (ssp245, 2041–2060) to 61.94 $\times$ 10$^4$ km$^2$

(ssp585, 2081–2100). In contrast, the size of the habitats in North America ranged from $33.58 \times 10^4$ km$^2$ (ssp245, 2041–2060) to $160.60 \times 10^4$ km$^2$ (ssp585, 2081–2100) (Table S4). North America has the largest area of suitable expansion. We observed that habitat expansion in Asia has occurred mainly in eastern Russia, eastern Kazakhstan, and eastern Afghanistan. The European habitat extension regions were mostly in western Russia, Finland, and Sweden, comparatively, the majority of the North American habitat extension regions were in the US and Canada. The study showed that the southern hemisphere's orchardgrass habitat is rapidly shrinking, with South American habitat areas declining by $16.74 \times 10^4$ km$^2$ (ssp126, 2041–2060) to $36.20 \times 10^4$ km$^2$ (ssp585, 2081–2100), which was 14.29 to 48.43 times the extent of expansion ($4.04 \times 10^3$ km$^2$ [ssp126, 2041–2060] to $2.53 \times 10^3$ km$^2$ [ssp585, 2081–2100]). On the other hand, Oceania contracted by $14.70 \times 10^4$ km$^2$ (ssp124, 2041–2060) to $28.22 \times 10^4$ km$^2$ (ssp245, 2081–2100), which was 4.01 to 24.29 times the extent of expansion ($1.16 \times 10^4$ km$^2$ [ssp245,2081–2100] to $3.94 \times 10^4$ km$^2$ [ssp126, 2081–2100]). African habitat areas contracted by $24.90 \times 10^4$ km$^2$ (ssp126, 2041–2060) to $33.57 \times 10^4$ km$^2$ (ssp126, 2081–2100), which was 0.80 to 5.70 times the extent of expansion ($4.37 \times 10^4$ [ssp126, 2041–2060] to $72.78 \times 10^4$ km$^2$ [ssp585, 2081–2100]) (Table S4). The majority of the contracted regions were in Congo, Morocco, and Libya in Africa, Argentina and Chile in South America, and southeastern Australia and New Zealand in Oceania.

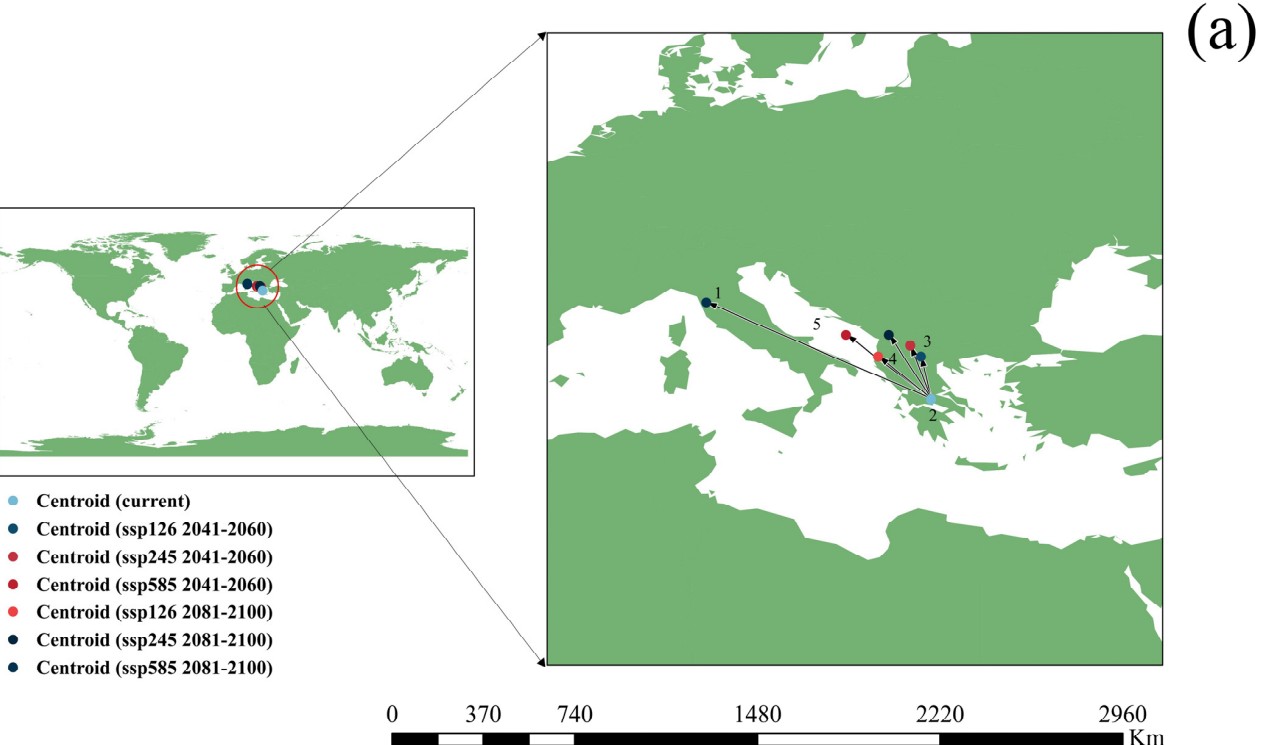

**Figure 6.** *Cont.*

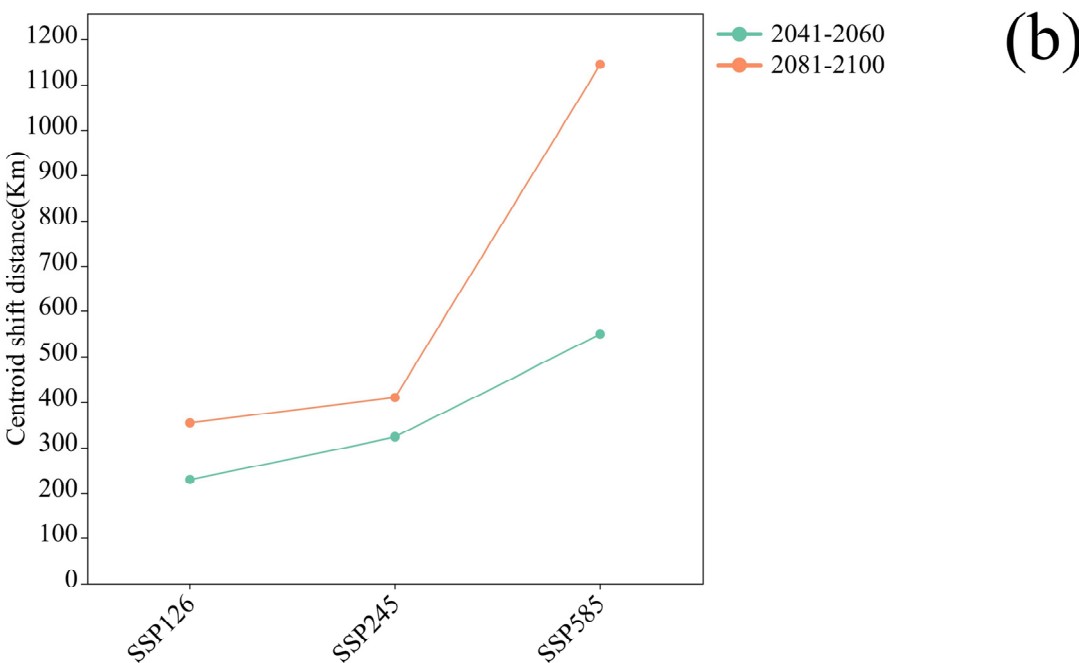

**Figure 6.** (**a**) Centroid changes for orchardgrass between current and future climate change scenarios. 1, Italy; 2, Greece; 3, Macedonia; 4, Republic of Albania; 5, Mediterranean Sea. (**b**) Centroid changes and Centroid shift distance under different GHG emission scenarios for orchardgrass.

## 4. Discussion

### 4.1. MaxEnt Modeling

MaxEnt, unlike other software programmes, can evaluate model performance using the under-the-curve (AUC) value of the receiver operating characteristic (ROC) curve [57], correct the sample deviation when obtaining data from a known distribution [58,59], and lessen spatial biases for the global distribution of species in the GBIF database [60]. AUC values for orchardgrass in this study, which were greater than the random AUC value of 0.50 and close to 1.00, demonstrated the reliability of the built-in model. Therefore, our model is able to accurately calculate the global distribution of orchardgrass habitat based on its performance.

Although the MaxEnt model possesses higher predictive performance and is widely used for predicting species distribution changes, it still has limitations such as high computational complexity, limited explanatory power, and susceptibility to the influence of low-resolution environmental data [61]. While climate and soil conditions are taken into account by the MaxEnt model, additional factors including adaptive ability, interspecific relationships, human development, and land utilization may also influence species distribution. In assessing model performance, AUC is sometimes not sufficient to assess model performance, and other metrics such as the Kappa coefficient and TSS are also important [62]. It is suggested that future studies may consider using more metrics to assess model performance. As a result, when all relevant factors are taken into account, we anticipate that orchardgrass suitable habitat will decline even more as a result of future global warming, and the reduction in habitat area may be more dramatic. However, the ensuing simulations might not always be more accurate if all variables are included in the model, because the impact of important variables might be reduced. Therefore, scientific selection of variables is needed to make the model more accurate and the results more meaningful.

### 4.2. Suitable Habitat Distribution Patterns of Orchardgrass under the Current Environment

Orchardgrass has a complicated distribution and diversity, most of which may be explained by its recent evolutionary, genomic histories, and migratory [63]. In general, response curves can be used to correlate species occurrence probability with major ecological factors. If a species is likely to have a probability of occurrence greater than 60%, the related ecological factor thresholds are appropriate for the existence of this species [26,64,65]. According to MaxEnt results and environmental factor response curves, the key environmental factors influencing distribution for orchardgrass in this study were temperature seasonality, mean diurnal range, maximum temperature of the warmest month, and precipitation of the coldest quarter. Findings from this study are consistent with those from previous studies [66,67].

According to our research, orchardgrass has a total suitable habitat area of $2133.01 \times 10^4$ km$^2$, with the majority of these areas being in Europe (including the Eastern European Plain, the Central European Plain, and the Western European Plain), Asia (including the Sichuan Basin, Yunnan-Guizhou Plateau, Northeast Plain, Middle and Lower Yangtze Plain, the North China Plain, and Japan), and North America (including the Cascade Mountain). This is consistent with the field distribution site of orchardgrass [68]. Since high-suitability locations are more beneficial to preserving orchardgrass variety than low-suitability areas, breeding, cultivation, and ex-situ conservation of orchardgrass species should be carried out in these locations.

### 4.3. Response of Suitable Habitats for Orchardgrass to Future Climate Change

Climate change, particularly worldwide warming, alters the pattern of precipitation distribution in addition to changing local temperatures. The distribution of these climatic factors will shift when the change is near to or over the adaptation threshold of the present plant development [69]. Our research demonstrates that different climate scenarios have different areas and distributions of orchardgrass suitable habitats, suggesting that the distribution of these habitats was impacted by climate change in spatially particular ways.

Among the 16 environmental variables adopted in the model, temperature seasonality and mean diurnal range made the greatest contributions to the distribution model for orchardgrass compared to other variables, indicating that these factors play important roles in its distribution. The probability of orchardgrass distribution increases and then decreases with the increase in temperature seasonality and mean diurnal range.

This result is supported by the fact that the climatic characteristics of an area act as key elements for population regeneration [70]. Orchardgrass is widely distributed in temperate and tropical regions of central and western Asia, temperate regions of southwestern Europe, and the Canary Islands of northern and western Africa [71]. Temperature seasonality refers to the magnitude of the temperature change between seasons [72]. In temperate and tropical regions of central and western Asia, where winters are cold and summers are hot, this temperature variation affects orchardgrass growth and dormancy cycles [73]. In the temperate regions of south-west Europe, where winters are warm and humid and summers are hot and dry, orchardgrass is well adapted to such environmental conditions [74,75]. This may explain the large values of temperature seasonality in orchardgrass. In contrast, in the Canary Islands in northern and western Africa, temperature seasonality is relatively small, and the warm and stable climate allows orchardgrass to grow and reproduce continuously. These connections help us better understand the ecological characteristics of orchardgrass and their interactions with the environment.

There is a certain relationship between plant growth and development and mean diurnal range [76]. The diurnal range is the difference between the daily maximum and minimum temperatures and is often used to describe the change in temperature between day and night [77]. Temperature regimes with a 5–15 °C amplitude enhanced seed germination percentages of orchardgrass, indicating that the conditional dormancy was released by these temperature regimes. Seeds germinated more rapidly under alternating temperatures than under constant temperatures. The dual effects of temperature on dormancy breaking

and germination were accounted for by thermal time models based on alternating temperature regimes [78]. The impact of different diurnal temperature ranges on the growth of orchardgrass varies. Orchardgrass grown in the 21 °C day/13 °C night environment produced more aerial dry matter and had a larger leaf area compared to orchardgrass grown in the 29 °C day/31 °C night environment [79]. Under warm conditions (32 °C day/24 °C night), orchardgrass produces fewer flowering stems, while under cool or temperate conditions (18 °C day/10 °C night), it produces a higher number of flowering stems. Additionally, under these temperature conditions, orchardgrass exhibits the highest levels of growth rate, yield, tillering, leaf area, and dry matter allocation [80]. Therefore, the growth and adaptability of orchardgrass may be influenced by the diurnal temperature range in its surrounding environment. Understanding this relationship helps to gain deeper insights into the ecological adaptability of goosegrass under different environmental conditions and can provide valuable references and guidance for agricultural applications.

Our findings suggest suitable habitats for orchardgrass will experience considerable expansion and contraction across continents according to future climate scenarios. Orchardgrass is highly resistant to cold but shows cessation of growth at extreme temperatures [66]. With global warming, some high latitudes in the northern hemisphere may become suitable habitats for orchardgrass due to increased temperatures, whereas some low latitudes may face extended periods of extreme heat or dryness, posing a threat to the survival of orchardgrass. orchardgrass distribution continually changes north-westward, with European areas at its center, under future climatic circumstances. As the temperature goes up, the orchardgrass distribution area shifts farther away. The offset distances under different GHG emission scenarios in 2041–2060 are smaller than those in 2081–2100, and they reach a maximum of 1143.49 km under the high concentration emission scenario in 2081–2100 (Figure 6b). As a result, it is already evident that places at high latitudes are becoming more appropriate for orchardgrass, but as global warming progresses, orchardgrass habitats will generally decrease. Due to projected global warming, the southern hemisphere may no longer be suitable for orchardgrass, including South Africa and Angola in Africa, Chile and Argentina in South America, and southern Australia and New Zealand in Oceania. Surveys and collection of orchardgrass germplasms in these places are required to conserve the genetic variety of this plant, and some exceptional or unusual germplasm can be maintained in vitro for future use by asexual propagation. Areas such as Europe, the hilly and mountainous regions in southwest China and the Yangtze River Basin, as well as the northeast region of China and the Cascade Mountain Range, and Pacific Coast Ranges, which are mostly unaffected by climate change can serve as a base for future large-scale orchardgrass cultivation and usage, the preservation of local genetic resources, and other agricultural activities.

## 5. Conclusions

A MaxEnt model was successfully developed to represent the currently suitable areas for orchardgrass and predict the potential distribution of orchardgrass under future climate scenarios. The temperature seasonality and mean diurnal range were determined in this study as the ecological thresholds for the important environmental variables. Based on the results of the study, there are important implications for governmental departments to formulate relevant policies: firstly, protection and management, through the establishment of nature reserves, prohibiting destructive development and other measures to protect the ecological environment of orchardgrass; secondly, promotion of large-scale production, through financial support, technical training and other means to help farmers to carry out sustainable production of orchardgrass; thirdly, the formulation of policies and standards, including land-use planning, environmental protection regulations; Thirdly, to formulate policies and standards, including land use planning and environmental protection regulations, to protect and utilize orchardgrass; and lastly, to support research and monitoring, and to provide a scientific basis for policy adjustments and preventive measures by funding scientific research and monitoring work. These insights will help to balance economic

development and ecological conservation goals to ensure the sustainable development of orchardgrass and the integrity of the ecosystem.

**Supplementary Materials:** The following supporting information can be downloaded at: https://www.mdpi.com/article/10.3390/agronomy13081985/s1, Table S1: The occurrence records of orchardgrass in the world; Table S2: Pearson correlation coefficient of environmental variables; Table S3: Projected potential suitable areas for orchardgrass under current and future climate scenarios; Table S4: The area change of suitable habitat of orchardgrass under different future climate scenarios on different continents; Figure S1: Receiver operating characteristic under future climate conditions; Figure S2: Response curves of the main predictors of orchardgrass citri occurrence probability. Curves show the mean response over ten replicate Maxent runs (red) and the mean ± 1 SD (blue, two shades for categorical variables).

**Author Contributions:** Methodology, J.W. and L.Y.; Software, J.W., L.Y., J.Z. and J.P.; Investigation, Y.X. (Yi Xiong); Resources, J.P., Y.X. (Yi Xiong) and Y.X. (Yanli Xiong); Data curation, J.W., L.Y., J.Z. and J.P.; Writing—original draft, J.W. and L.Y.; Writing—review & editing, X.M.; Visualization, J.W. and L.Y.; Supervision, X.M.; Funding acquisition, X.M. All authors have read and agreed to the published version of the manuscript.

**Funding:** This project was supported by the Sichuan Province "14th Five-Year Plan" Forage Breeding Research Project (2021YFYZ0013-2), Beef Innovation Team (SCCXTD-20-13), the National Modern Forages Industry Technology System (CARS-34) and Sichuan Forage Innovation Team Program (SCCXTD-2020-16).

**Data Availability Statement:** Please contact the first author with requests for data.

**Conflicts of Interest:** The authors declare no conflict of interest.

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
