# Peer review of "Modeling Climate Change Indicates Potential Shifts in the Global Distribution of Orchardgrass"

_agronomy, doi:10.3390/agronomy13081985_

Round 1
Reviewer 1 Report
Please see the attachment.

Reviewer 2 Report
The aim of the paper is to model the potential shifts in the global distribution of orchardgrass under climate change. The main contribution of the paper is the use of a modeling approach to predict the potential changes in the distribution of orchardgrass, which is an important forage crop. The paper's strengths include a detailed description of the methods used, a clear presentation of the results, and a discussion of the implications of the findings for the management of orchardgrass in the face of climate change.
Reviewer 3 Report
The manuscript is about the species distribution modeling of orchardgrass where authors implemented the recent package of R-Kuenm for SDM. Methodology, results and discussion section of the manuscript need some improvement. I have some suggestions for the improvement of the manuscript in the attached file.

Moderate language improvement needed
Round 2
Reviewer 1 Report
The manuscript is sufficiently improved in the revised version.
Best wishes,